# Sampling Error and Its Implication for Capture Fisheries Management in Ghana

**Temiloluwa Jesutofunmi Akinyemi** [1,2,*], **Denis Worlanyo Aheto** [1,2] **and Wisdom Akpalu** [3]

1   Centre for Coastal Management, Africa Centre of Excellence in Coastal Resilience (ACECoR),
    University of Cape Coast, Cape Coast PMB TF0494, Ghana
2   Department of Fisheries and Aquatic Sciences, School of Biological Sciences, College of Agriculture and
    Natural Sciences, University of Cape of Coast, Cape Coast PMB TF0494, Ghana
3   School of Research and Graduate Studies, Ghana Institute of Management and Public
    Administration (GIMPA), Accra 039 5028, Ghana
*   Correspondence: temiloluwa.akinyemi@stu.ucc.edu.gh; Tel.: +233-(0)-2-4857-4617

**Abstract:** Capture fisheries in developing countries provide significant animal protein and directly supports the livelihoods of several communities. However, the misperception of biophysical dynamics owing to a lack of adequate scientific data has contributed to the suboptimal management in marine capture fisheries. This is because yield and catch potentials are sensitive to the quality of catch and effort data. Yet, studies on fisheries data collection practices in developing countries are hard to find. This study investigates the data collection methods utilized by fisheries technical officers within the four fishing regions of Ghana. We found that the officers employed data collection and sampling procedures which were not consistent with the technical guidelines curated by FAO. For example, 50 instead of 166 landing sites were sampled, while 290 instead of 372 canoes were sampled. We argue that such sampling errors could result in the over-capitalization of capture fish stocks and significant losses in resource rents.

**Keywords:** fisheries data quality; fisheries management; Ghana





## 1. Introduction

Millions of people around the world depend, directly or indirectly, on capture fisheries for their food security, income, and livelihoods [1]. This dependence is particularly strong in coastal communities in developing countries where the sector employs 97% of the 50 million people who make up the world's fishing workforce [2,3]. Ghana, as one of the developing countries, is home to a wide variety of biodiversity, including small pelagic species such as anchovies, sardinella, and chub mackerel and larger pelagic fish such as yellowfin, skipjack, and big-eye tuna. There are also demersal fish such as grouper and snapper, and other seafood such as shrimp and squids [4]. For sustenance and the eradication of poverty, a majority of coastal dwellers are solely dependent on the exploitation of these fisheries, with over 60% of the population relying on fish as their primary source of protein and about 10% of the population (2.6 million people out of a total population of 26 million) believed to be directly or indirectly dependent on fish resources [5].

The Ghanaian marine fishing industry is divided into three primary sectors: small-scale artisanal fishers, semi-industrial fisheries, and large industrial fisheries, with over 300 fish landing sites spread throughout its coasts [6]. The vessels used in Ghana's marine capture fishery include dugout canoes, canoes with outboard motors, trawlers, and large steel-hulled foreign-built vessels. The dugout canoes and canoes fitted with outboard motors are primarily utilized by artisanal fishers while trawlers and steel-hulled vessels are used mainly in the semi-industrial and industrial marine fisheries [7]. There is currently a total of 11,583 licensed marine artisanal canoes operating along the coast, 150 semi-industrial vessels, and 84 licensed industrial trawlers in Ghana's marine waters [8].

Despite the importance of the fisheries sectors, according to various experts, fish stocks have been declining rapidly due to the overcapacity of fleets, excessive fishing quotas, illegal fishing practices, and the generally poor management of fisheries, which poses existential threats to coastal communities [9]. This has necessitated the formulation and refinement of existing management policies with the aim of limiting fishing efforts to optimize the economic, social, and ecological sustainability of capture fisheries [10]. The effectiveness of effort-limiting policies, however, depends on the availability and quality of the relevant fisheries data used for decision-making [11].

National governments and international organizations have been working hard at collecting fisheries data to inform sustainable and long-lasting management plans and strategies [12]. However, this remains a daunting task due to the complex interactions among species and marine ecosystems, and the wide distribution and migration of pelagic stocks across national jurisdictions. These complexities of biophysical dynamics make fisheries management difficult [13]. Nevertheless, management decisions must be made as livelihoods and incomes depend on wise decisions made by the managers, and they can only make wise decisions if they have sufficient knowledge of the ecosystem and fishery to understand the causes of the current fisheries situation and predict how the resource and fishery will change in response to management actions [14].

Accurate and consistent knowledge about how a fishery is doing, as well as what, where, and how much of a species is being captured requires more precise data collection and faster and more advanced reporting, processing, and analysis, as well as more efficient mechanisms to disseminate the results to enable close to real-time analysis [15]. The fisheries data collected is usually the manager's major source of information, which is essential in developing appropriate management tools to support the sustainable use of the stock [16,17]. However, the data quality is low in many developing countries owing to inadequate resources, including skills and funding.

Although the FAO Code of Conduct (Paragraph 6.4) has stated that the conservation and management of fisheries must be based on the best scientific knowledge available at any point in time. Unfortunately, many fisheries agencies lack sufficient data, making attempts at managing fisheries difficult. For instance, the reconstruction of catches carried out by [18,19] revealed that the catch and effort data compiled by FAO were deficient. As noted by [20], the unavailability and suspicion of errors in catch data due to lack of skills and resources in member countries have resulted in the complementation or replacement of countries' data with data from other sources. These omissions or errors in data collection could lead to erroneous fisheries management policies, which in turn could result in suboptimal extraction, losses in resource rents, and eventual collapse of capture fisheries. It is therefore expedient to assess how catch and effort data are collected to better inform management policies.

An analysis of the national fisheries data collection protocols in Ghana suggests that the Fisheries Scientific Survey Division (FSSD) is mandated to conduct scientific research and deploy surveys on marine environments and fisheries to inform the formulation and management of policies aimed at the sustainable management of Ghana's marine fisheries resources. The FSSD is under the Fisheries Commission (FC), which was established in 1962 with technical assistance from the Food and Agriculture Organization (FAO). Due to limited human and financial resources, the FSSD has not been able to provide adequate monitoring of the data collection activities of the technical officers. Thus, any errors that occur on the field are ignored. To the best of our knowledge, no study has been undertaken to investigate whether the recommended sampling procedures are followed by the field enumerators.

## 2. Materials and Methods

### 2.1. Study Area

This study was carried out in the following twenty-nine fish landing sites out of thirty representing the four coastal administrative regions in Ghana: Abutiakope, Lighthouse (Volta), Gbegbeyise, Botianor, Agjivompanye, Odin-nyonma, Osu alata, Teshie, Ga mashie,

Awudun (Greater Accra), Saltpond, Kromantse, Apam main, Elmina main, Elmina, Ayipey, Abrofo mpoano, Mumford main, Enfano (Central Region), Dixcove, Sekondi, Fante line (Axim), Akyinim, Ewe line, Fante line (Half Assini), Sharma Apo, Sekondi-Takoradi, Akwadae, and Adjua (Western) (See Figure 1 for the geographical location of the landing sites). These twenty-nine sites chosen for primary data collection were selected based on the total number of enumerators in Ghana and where they are assigned along the coast.

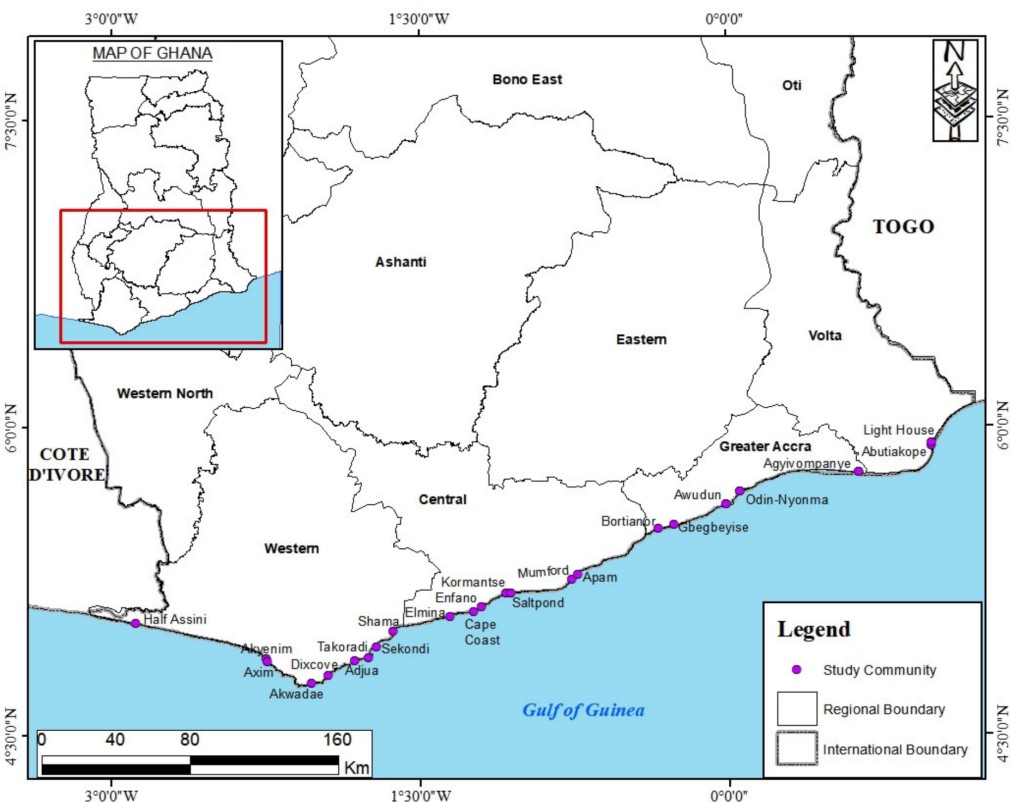

**Figure 1.** Map showing the twenty-nine landing beaches in Ghana.

### 2.2. Research Design

This study used a quantitative survey design to examine the types of data collected and the methods associated with the data collection in Ghana. The data were gathered between May and June 2022 at all 30 landing sites. A structured questionnaire was used for data collection. Field assistants were trained on the administration of the questionnaire, ethical standards, and COVID-19 safety protocols. The respondents included 29 Fisheries Commission field enumerators and 1 Field Scientific Survey Division data manager. The surveys were conducted in English and local languages, including Fante, Ga, Nzema, and Twi. Each interview lasted between 40 to 60 min.

### 2.3. Research Instrument

The questionnaire used was made up of three sections. The first part of the questionnaire (Section A) consisted of an introductory statement and questions about the relevant sociodemographic characteristics of respondents. Some of the variables included age, years of experience in data collection, the number of landing sites, gender, fishing experience, and level of education (basic education, secondary, and tertiary). The next two sections highlighted the types of fisheries data gathered using the FAO data collection guidance as a benchmark [21]. The data was classified as biological, ecological, economic, or social. A total of 24 questions were developed through an extensive review of the literature [14,16,22,23]. For the evaluation of the data collection procedure, the questions comprised five categories. These categories were based on the source of the data on fish production, the type of effort data gathered, the type of capture data gathered, and the frequency of data collection.

*2.4. Data Analysis*

Responses from interviews were coded using the IBM Statistical Package for Social Scientists (SPSS) computer software version 20.0. (2012) and analyzed for trends in response to research questions using Software for Statistical Analysis (STATA SE 15.0) (STATA Corp, College Station, TX, USA) and Microsoft excel. To understand the distributions of all relevant variables, descriptive statistics (frequencies and percentages) were generated. The summaries of the results are presented in tables (Tables 1–3). To check for sampling error, this study compared the capture fisheries data collection procedures in Ghana to the recommended best practices (i.e., the FAO guidelines) along the entire coast of Ghana using the FAO toolkit for small-scale fisheries routine data collection [22] and the FAO data collection guidelines [21]. The sample size formula developed by [24] was used to estimate the actual sample size for comparison with the number sampled.

**Table 1.** Catch data collection by Ghana's Fisheries Commission enumerators.

| Variables | Collect (%) | Do Not Collect (%) |
|---|---|---|
| **Biological data** | | |
| Total fish landings by major species | 66 | 34 |
| Total fish landings by canoes | 69 | 31 |
| The total effort by canoes | 86 | 14 |
| Length and/or age composition of fish landings | 21 | 79 |
| Discards of fish species per canoe | 0 | 100 |
| Length and/or age composition of discards | 0 | 100 |
| Areas fished by each canoe | 17 | 83 |
| **Ecological data** | | |
| Total catches of bycatch species | 17 | 83 |
| Length and/or age composition of bycatch | 3 | 97 |
| **Economic data** | | |
| The average income per fishing unit | 52 | 48 |
| The cost of premix fuel | 7 | 93 |
| Price of fish landed per canoe | 93 | 7 |
| **Social data** | | |
| Crew size within each canoe | 93 | 7 |

**Table 2.** Type and method of catch data collected by enumerators.

| Variables | Freq | Percent |
|---|---|---|
| **Type of catch data collected** | | |
| Multi-species (all species) | 15 | 51.72 |
| Single-species (only one species) | 10 | 34.48 |
| Single-species and multi-species | 4 | 13.79 |
| **Data collection method** | | |
| By canoes | 10 | 34.48 |
| By gear | 5 | 17.24 |
| By species | 10 | 34.48 |
| By species and gear | 4 | 13.79 |

To enhance visualization and appreciation of the study context, graphs are presented. The landing sites and canoes sampled across the whole district were compared with the landing sites and canoes that were required to be sampled. Summaries of the results are presented in Figures 2–5. The chi-square test was then used to verify whether there was a significant difference between the actual and the expected sampled landing beaches.

**Table 3.** Type of effort data collected by enumerators in each district (✔ = data collected; ✘ = Data not collected).

| District | Number of Canoes | Size of Fishing Gear | Type of Fishing Gear | Number of Trips | Trip Duration | Size of Canoe |
|---|---|---|---|---|---|---|
| Keta | ✔ | ✘ | ✔ | ✘ | ✔ | ✘ |
| Ada East | ✔ | ✔ | ✔ | ✘ | ✔ | ✘ |
| Kpone Ketamanso | ✔ | ✘ | ✔ | ✘ | ✔ | ✘ |
| AMA | ✔ | ✔ | ✔ | ✔ | ✔ | ✘ |
| TMA | ✔ | ✘ | ✔ | ✘ | ✔ | ✘ |
| Ga South | ✔ | ✘ | ✔ | ✔ | ✔ | ✘ |
| Efutu Municipal | ✔ | ✘ | ✔ | ✔ | ✔ | ✘ |
| Gomoa West | ✔ | ✘ | ✔ | ✔ | ✔ | ✘ |
| Ahanta West | ✔ | ✘ | ✔ | ✔ | ✔ | ✘ |
| Abura-Asebu Kwamankes | ✘ | ✘ | ✘ | ✔ | ✔ | ✘ |
| Cape Coast | ✘ | ✘ | ✔ | ✔ | ✔ | ✘ |
| Nzema East | ✔ | ✘ | ✔ | ✘ | ✔ | ✘ |
| Jomoro | ✔ | ✘ | ✔ | ✔ | ✔ | ✘ |
| Komenda-Edina-Equafo | ✔ | ✘ | ✔ | ✘ | ✔ | ✘ |
| Ledzokuku | ✔ | ✔ | ✘ | ✔ | ✔ | ✔ |
| Mfantseman | ✔ | ✘ | ✔ | ✔ | ✔ | ✘ |
| Sekondi-Takoradi | ✔ | ✘ | ✔ | ✔ | ✔ | ✘ |
| Shama | ✔ | ✘ | ✔ | ✘ | ✔ | ✘ |

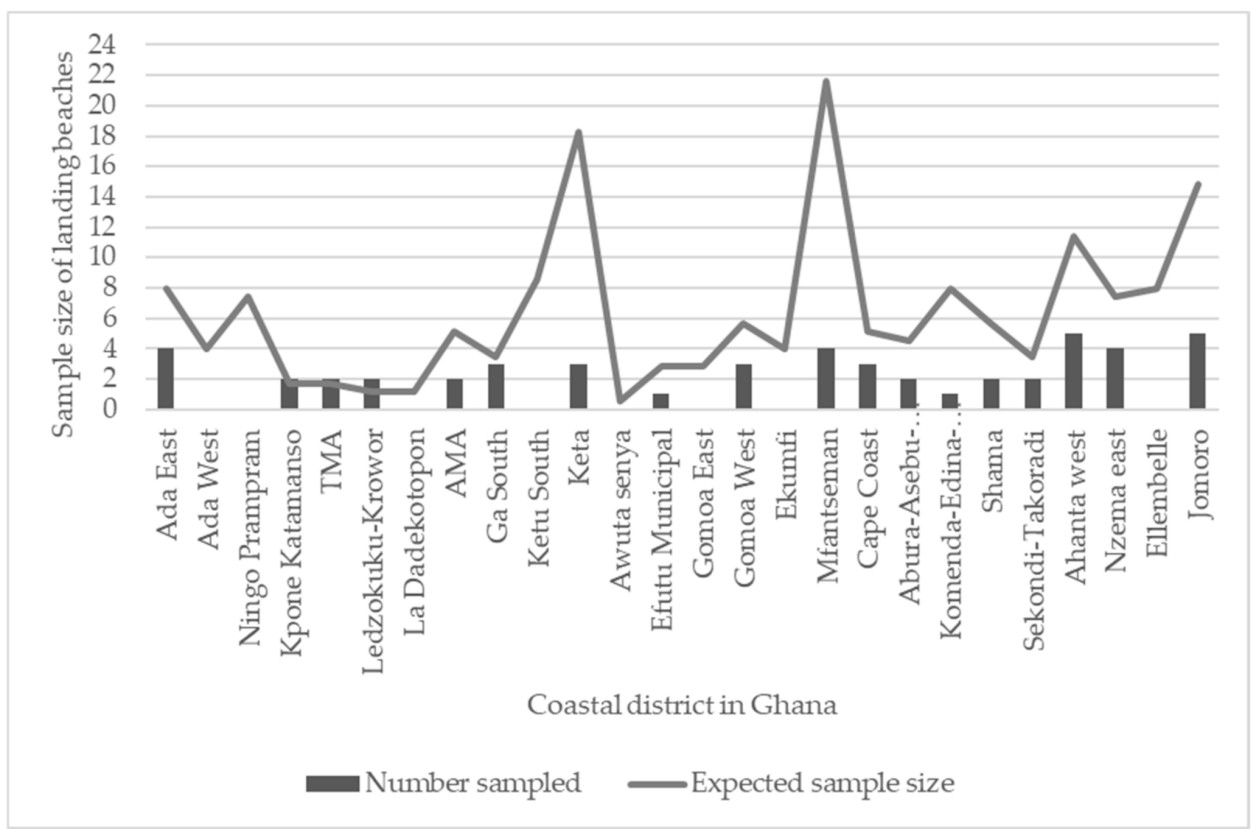

**Figure 2.** Comparison between the observed sample size of the Fisheries Scientific Survey Division (FSSD) and the expected sample size of landing beaches according to the districts in Ghana.

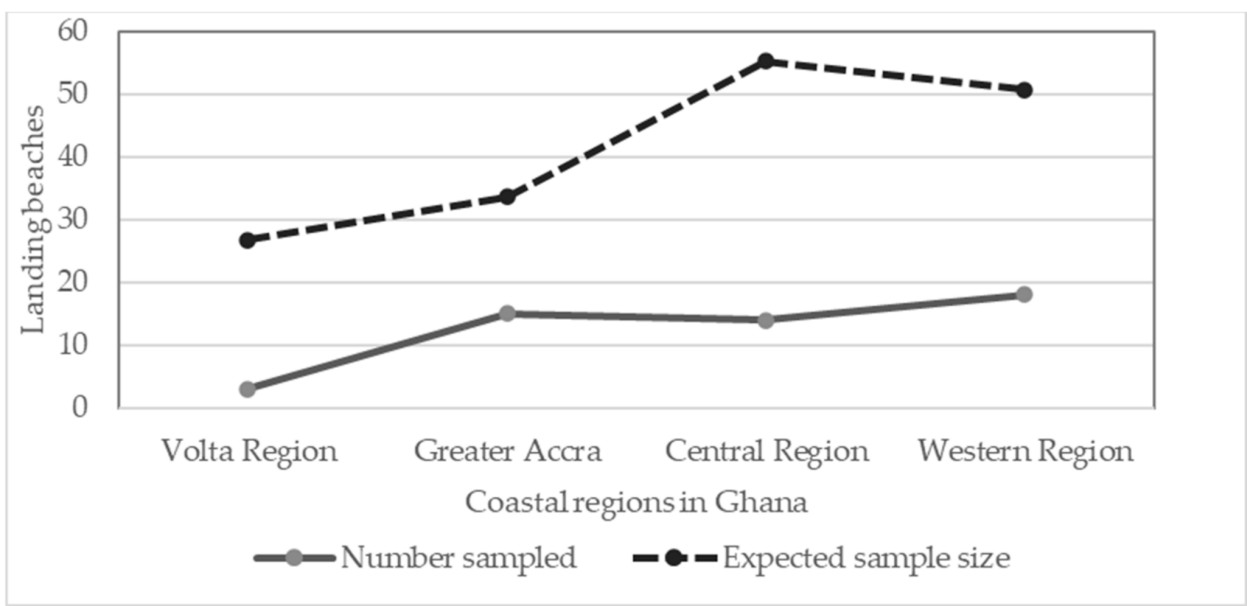

**Figure 3.** Comparison between the observed sample size of the Fisheries Scientific Survey Division (FSSD) and the expected sample size of landing beaches according to the coastal regions in Ghana.

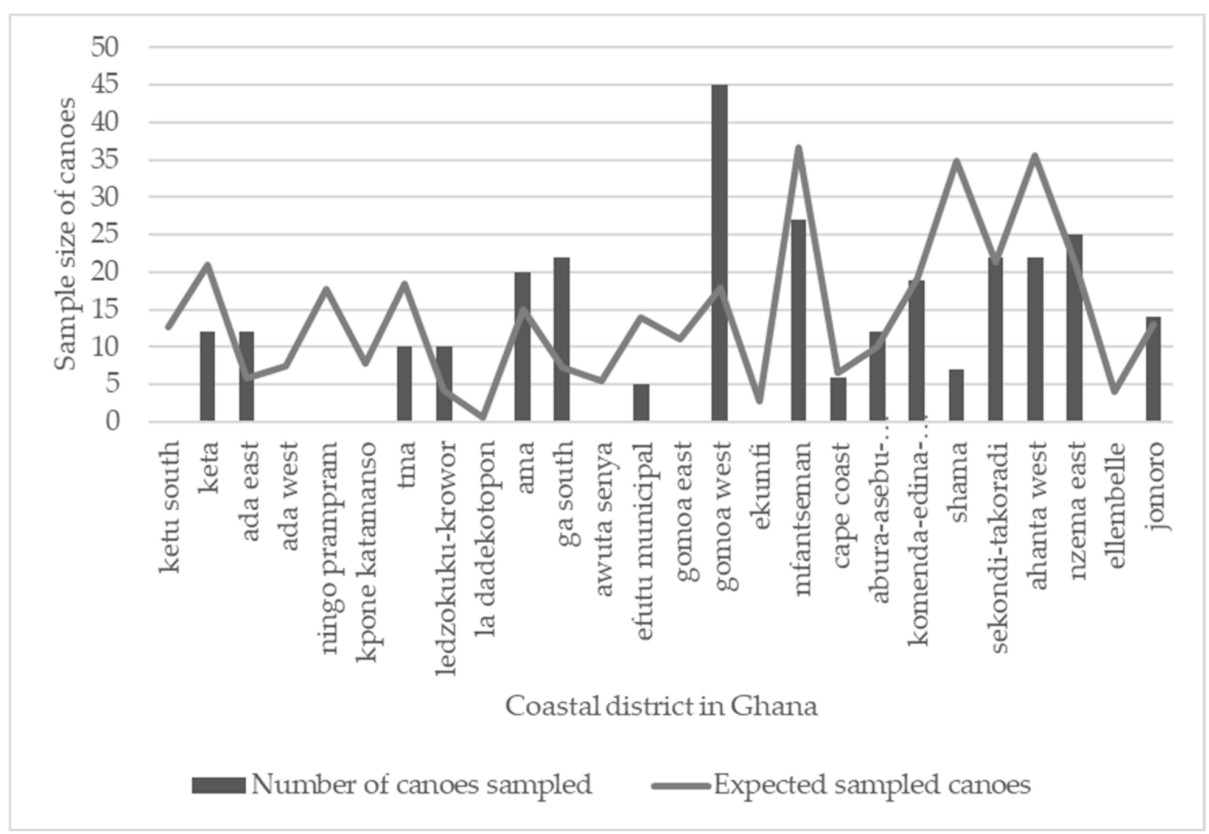

**Figure 4.** Comparison between the observed sample size of the Fisheries Scientific Survey Division (FSSD) and the expected sample size of canoes according to the districts in Ghana.

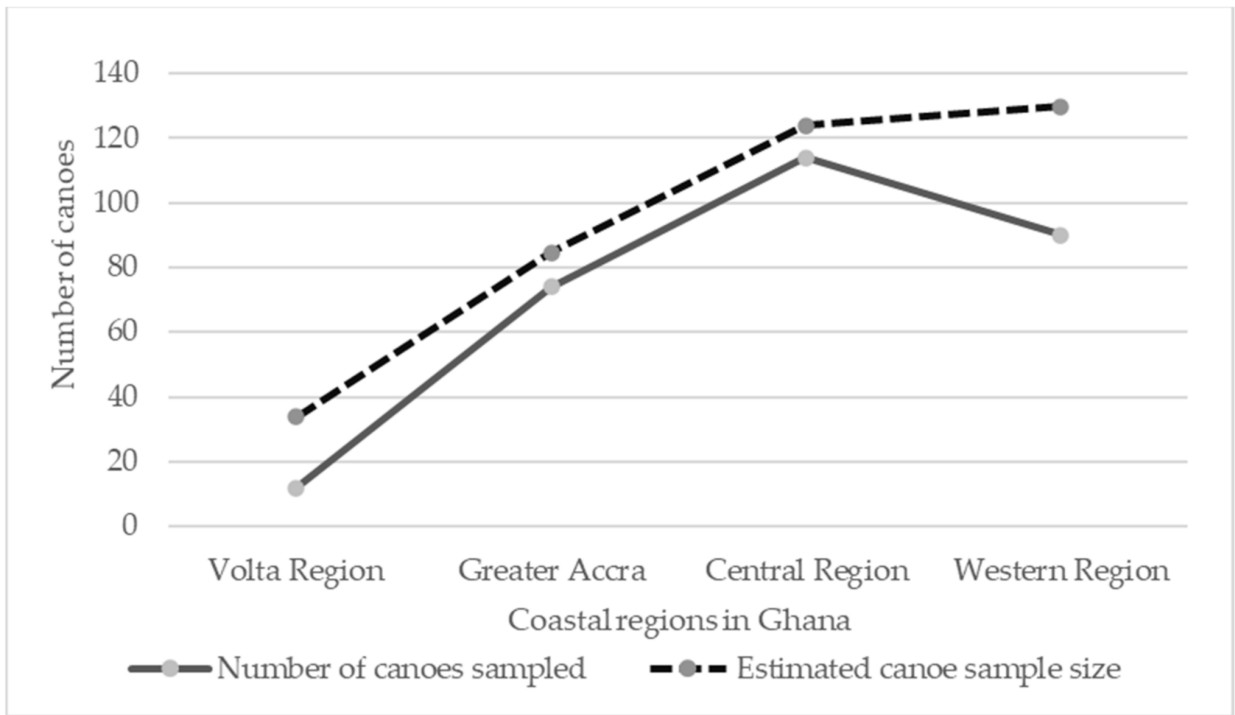

**Figure 5.** Comparison between the observed sample size and the expected sample size of canoes according to the coastal regions in Ghana.

### 3. Results

This section presents the data collection procedures of the FSSD and the results of the study. FSSD employs 30 enumerators to collect artisanal fisheries data from 50 landing beaches out of approximately 292 landing beaches [8]. These 50 landing beaches were obtained using the three-stage sampling survey by dividing the whole coastal area into four regions (i.e., Major strata) and the four regions into districts (Minor strata). Sampled canoes and landing sites were selected within the minor strata (districts) based on the canoe frame survey for the sole purpose of increasing the accuracy of the derived estimates using the proportional stratified sampling method.

The equation for the sampling is $n_k = \frac{n}{N} \times N_k$, with a maximum of 12 canoes sampled daily, where N and n are the total population and sample sizes, respectively, k is the number of strata, $N_k$ is the number of units in stratum k, and $n_k$ is the number of sampled units in stratum k. To calculate the sample size of the total population, sample size formula $n = \frac{N}{1 + N(e^2)}$ is used [24], where e is the level of precision. The FSSD employ 95% as the confidence interval and +/−5 as the degree of accuracy.

The sampling procedure is adopted from the FAO toolkit for small-scale fisheries routine data collection [22] and, as stated in the toolkit, enumerators at the landing beaches sample data for 14 days/gear/month with each enumerator having two gears which in some cases spill over to two landing sites each, depending on the size of the landing site and the abundance of fishing gears. For the recording of data, two forms are provided by the Fisheries Commission, Forms 1a and 1b, with each performing a different function. Form 1a is used to record daily information on fishing activity at the landing site, and Form 1b is used to record information and data collected. The data collected by FSSD are placed into three categories: the fisheries statistical data, i.e., the catch and effort data; the biological data; and the environmental data.

The fisheries data expected to be recorded at each landing beach are catch and species composition (single-species), fishing effort, price of fish, number of operating fishing crafts, types and sizes of fishing crafts, types of gears and their target species, areas of operation of fishing crafts, number of fishermen on fishing crafts, and information on landing sites.

The biological data to be collected are fish length, fish weight, gonad weight, and sex. The environmental data are salinity, temperature, and dissolved oxygen.

The data is collected by the 30 enumerators at all fifty landing sites but due to the shortage of resources and intellectual capacity, the biological data is collected at four landing sites across the four coastal regions, and the environmental data is collected at six landing sites. These sites were selected based on the abundance of fish species and the flow of water, respectively. After the data are recorded, the zonal officials (supervisors) in charge of the enumerators receive the records from each landing site and transmit them to FSSD, where they are compiled. For this study, since 1 of the enumerators had hearing loss, only 29 were interviewed. These enumerators had a male-to-female ratio of 25:4, an average age of 37 years, and an average of 10 years of data collection experience.

### 3.1. Sampling of Landing Beaches in the Coastal Districts of Ghana

Ghana has 292 landing beaches. This means that the calculated sample size is 166 [24]. However, only about a third of these beaches are sampled by enumerators. Each of the 26 fishing districts should have at least 1 landing beach sampled based on ratio and proportion, but as can be seen in Figure 2, the enumerators cover 18 out of the 26 fishing districts, resulting in an under-sampling of 8 districts. The data gatherers also stated that they sample a total of 50 landing beaches from the 18 districts they work in, which is 80 beaches less than what should be sampled from those 18 districts (assuming the sampling of the 18 districts is desirable). However, they over-sample in Ledzokuku-Krowor by 1 landing beach.

To determine whether there is a significant difference between the number of landing beaches sampled and the number of landing beaches expected to be sampled, a chi-square test was undertaken, and we found a significant difference (93.87276, *p*-value of 0.001). The low coverage of landing beaches is attributed to a lack of human and financial resources.

### 3.2. Sampling of Landing Beaches in the Coastal Regions of Ghana

On a regional level, we discovered a considerable discrepancy between the actual and expected landing beaches sampled, as shown graphically in Figure 3. This was found using the same methodology (sampling, ratio, and proportion). We discovered that the Central Region has a more pronounced under-sampling of 41 landing beaches as compared to Greater Accra which is under-sampled by 19 landing beaches.

### 3.3. Sampling of Canoes in the District of Ghana

Ghana had 11,583 canoes in total as of 2016, according to MoFAD. Out of this total, 372 canoes were to be sampled. Based on a proper sampling procedure, at least 1 canoe should be sampled from each coastal district. We also discovered from our research that the 290 canoes from 18 districts that the enumerators collectively sample are either under- or over-sampled. Figure 4 indicates that canoes are over-sampled in approximately half of the district, with Gomoa West and Ga South oversampled by 27 and 15 canoes, respectively. However, the canoes were under-sampled by 28 and 14 canoes in Sharma and Ahanta West districts, respectively. The over-sampling of canoes was found to be attributable to misalignment of incentives: i.e., compensation for the district from which data is not being collected or the district with a smaller number of canoes.

### 3.4. Sampling of Canoes in the Coastal Regions of Ghana

As presented in Figure 5, there are variations in the number of canoes sampled and the expected sample in each region. Clearly, there is under-sampling, with the Western region having the highest proportion of under-sampled canoes (40 canoes) as opposed to the other regions, especially the Central region which is under-sampled by 10 canoes.

### 3.5. Percentage Contributions

Each coastal region's contribution to the under-sampling of canoes and landing beaches is shown in Figure 6. Using the differences between the actual sampled with the estimated sample across the coastal regions, we found that the Western region contributed the most to the under-sampling of canoes (49%) and slightly less than the Central region to the under-sampling of landing beaches (28%), with the Central region contributing the most to the under-sampling of landing beaches (35%) and the least to the under-sampling of canoes (12%).

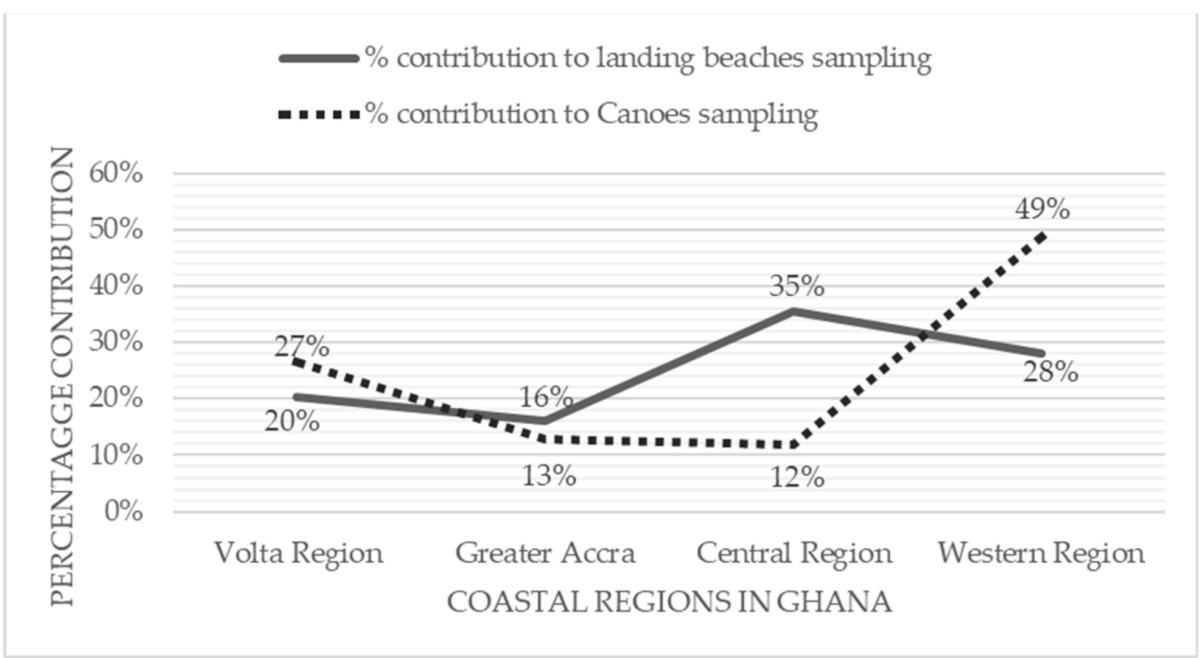

**Figure 6.** Percentage contribution of coastal regions to under-sampling of Canoes and Landing beaches in Ghana.

### 3.6. Catch Data

The four catch data categorizations (i.e., biological, ecological, economic, and social) were analyzed [14]. Each category has various components as presented in the first column of Table 1. The results revealed that none of the components under each thematic area show 100% data collection among the enumerators.

Out of the seven components under the biological category, the enumerators do not collect data on discards of fish species per canoe and the length and/or age composition of discards. However, 86% of the enumerators collect information on fishing effort. Only 17% of them indicated that they collect information on bycatch species, while 93% collect data on the price of fish and crew size of each canoe. These inconsistencies discount the reliability of the data aggregated by the FSSD for effective fisheries management.

Some of the enumerators collected data on single species and others on multiple species. The multi-species and single-species data indicate an ecosystem-based approach and a precautionary approach, respectively [25]. Due to the establishment of an ecosystem-based approach in national and international law, the authors of [26,27] suggested an ecosystem-based approach as the appropriate starting point for management; however, as indicated in Table 2, we can establish that there has been no consensus on which approach to use. About 52% of the enumerators collect multi-species (i.e., collect data by canoes and by gear) while 14% collect both single-species and multi-species (i.e., collect by either species or by both). The Fisheries Commission, on the other hand, indicated that the enumerators were instructed to collect only single-species data.

*3.7. Effort Data*

Regarding data on fishing effort, differences in the frequency across the enumerators were found. Comparing results from the 50 landing sites (Table 3), 86% of the enumerators collect data on the number of canoes and the type of gear, while only 16% indicated that they collect data on the size of a canoe.

## 4. Conclusions

From the survey on data collection practices by technical officers at the landing beaches, evidence of under-sampling and over-sampling has been found. This implies that the FAO toolkit for best practice is not being followed in practice in Ghana. This may be due to a lack of financial resources and the requisite skills to follow the desired protocol for fishery data collection. The sampling procedure deviates significantly from the ideal, which has implications for the quality of data generated.

A sample size that is too small might result in a Type I error [28], which is the likelihood of incorrectly rejecting a certain discovery when it should be accepted. Additionally, the author argued that an excessively high sample size is not appropriate due to the potential for type II error, which involves accepting a certain finding when it should be rejected. Thus, the relevant data needed for the formulation of management policies could be erroneous, thereby affecting the accuracy of the estimated catch and effort data.

The collection of catch and effort data sets and the method by which they are collected were different at some landing beaches. This discrepancy contrasts with FSSD's objective of collecting reliable data guided by scientific procedures. As noted by the authors [29], components of each thematic area should be the same at every landing site (beach) to ensure accurate data for fisheries management.

Errors in the sampling of landing beaches and canoes, as well as discrepancies in data sets gathered, could lead to the exaggeration of catch potentials, resulting in erroneous estimates of the maximum sustainable yield level (MSY) and the effort corresponding to maximum sustainable yield ($F_{MSY}$). These wrong estimates could lead to over-exploitation or over-capitalization of fisheries and their eventual collapse, as suggested by many studies.

To improve the quality of data collection, proper monitoring of the field enumerators should be incorporated as part of the Ministry's activities and the use of the FAO Open Data Kit (ODK) mobile phone application should be reviewed, upgraded, and its usage continued to ensure accurate collection of data. National service personnel from fisheries academic departments should also be employed to ensure better coverage of landing sites in the country. This suggestion comes with limited cost implications. In addition, there should be a balance between an understanding of the sampling techniques, the need for data, and the kind of data to be collected by the field enumerators and office staff.

**Author Contributions:** T.J.A.: Conceptualization, Methodology, Formal Analysis, Investigation/Field Data Collection, Writing—Original Draft, Project Administration. D.W.A.: Conceptualization, Validation, Funding, Writing—Review and Editing, Supervision. W.A.: Conceptualization, Validation, Formal Analysis, Writing—Review and Editing, Supervision. All authors have read and agreed to the published version of the manuscript.

**Funding:** This work was supported by the World Bank through the Africa Centre of Excellence in Coastal Resilience (ACECoR) Project at the Centre for Coastal Management, University of Cape Coast [Credit No.: 6389-GH].

**Institutional Review Board Statement:** As a result of the study's lack of danger, the study's sample size, and the Ministry of Fisheries and Aquaculture's consent that its staffs participate in it, ethical review and approval were waived for this study.

**Informed Consent Statement:** A letter was written to the Ministry of Fisheries and Aquaculture Development and to the technical officers to inform and obtain consent and consent has been obtained from them to publish this paper.

**Data Availability Statement:** All data herein are publicly available.

**Acknowledgments:** The authors would like to thank the World Bank for funding this research. We are also indebted to the Fisheries Survey Scientific Division staff and their field enumerators who dedicated their time to contribute to the research interviews. Finally, we wish to express our sincere gratitude to Mark Senanu Kudzordzi for supporting various aspects of the research.

**Conflicts of Interest:** The authors declare no conflict of interest.

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
