# Peer review of "Sampling Error and Its Implication for Capture Fisheries Management in Ghana"

_fishes, doi:10.3390/fishes7060333_

Round 1

Reviewer 1 Report

General comments:

The manuscript is actual and valuable. Every piece of information, concerning international fisheries and related problems, should be public, in order to be taken into account, so as to undertake future strategies. Nevertheless, certain information is here put down, somehow without proper ordering as in an accepted scientific article. This can be improved without much effort by the authors. The volume is relatively big, and the information included in the figures is again mentioned in the text. Materials and methods should be shorter and more precise. The results could be more robust, and additional synthesis on available data could be performed. Similar cases for valuable and appropriate comparisons to discuss are lacking, e.g., other neighboring countries, EU, USA and/or other FAO common fisheries policies, etc. This additional information could strengthen the manuscript, to obtain more scientific soundness.

Special comments:

Lines 69-71: Here results have been added in the introduction. This sentence is irrelevant to this chapter.

Lines 73-77: It is probably better to move this text to the M+M 3.2; 3.3 subchapters.

Chapter 2 is unnecessary:

Lines 79-84 are more relevant in the introduction.

The remaining text (lines 84-124) contains part of the methodology and has to be moved to the M+M chapter.

Line 127-134. This information is included in fig. 1, there is no need to be repeated in the text.

The legend of fig. 1 should be more informative.

Lines 167-173: This is explanatory text concerning results. It has to be rejected.

Line 178: what is the meaning of “experience level of 10 years”? Exactly 10, more or less?

The legend of fig.2 should be more informative: what kind of test is used for the comparison?

Fig. 3 and others as above.

Line 291: Are established “errors” only connected with undersampling or other issues were discovered during the survey? This is relevant to all other similar statements.

Conclusion:

The manuscript will be appropriate for publishing after a certain and obligatory review.

Reviewer 2 Report

Overall, this paper does a great job highlighting deficiencies in the current sampling scheme for coastal fisheries in Ghana. I have only minor edits to suggest, but also offer two higher-level general comments that might help make it an even stronger paper. These general suggestions include:

1.       The introduction could benefit from overview of a few sentences or so about Ghanan coastal fisheries, in terms of target species, stock structure, fishing methods, fishery management approaches, etc.—basically just enough info to give the reader sufficient understanding of why a robust and consistently applied coast-wide sampling program is necessary for sound management.

2.       The conclusion section could benefit from a few sentences on recommendations for strengthening the program going forward. Lines 297-300 do this to some extent, but are there ways in which additional resources (= more staff? new technology) or regular training regimes might tighten up the gap between target sampling levels and what’s achieved? If there’s any chance that this paper could help inspire such changes or leverage additional resources, then it would be worth highlighting a few such recommendations.

My list of minor editorial items includes the following:

-          Line 191 – is P = 0.311 correct? The statement (and chi-square value) imply significance, but this is far greater than the typically 0.05 significance level.

-          Line 207 – Relying on 2016 fleet size info for a 2022 survey – is there any chance that a now-several-year-old data set might contribute to inaccuracies in the expected sample size? Assumedly not, but it may be worth nothing this just in case other readers wonder.

-          Figure 6 – Suggest shifting the overlaid percentage values so they are legible, particularly those appearing on top of the darker gray line.

-          Line 254 – Reference to Daniel H and Sam S 2019 seems a bit odd and doesn’t appear in the citations, is there an error here?
